# The Role of Non-Neuronal Acetylcholine in the Autoimmune Blistering Disease Pemphigus Vulgaris

**DOI:** 10.3390/biology12030354

**Published:** 2023-02-23

**Authors:** Delila Pouldar Foulad, Nicola Cirillo, Sergei A. Grando

**Affiliations:** 1Division of Dermatology, University of California, Los Angeles, CA 90095, USA; 2Melbourne Dental School, University of Melbourne, Carlton, VI 3053, Australia; 3Department of Dermatology, University of California, Irvine, CA 92697, USA

**Keywords:** pemphigus, acetylcholine, keratinocyte adhesion, acantholysis, nicotinic receptor, muscarinic receptor, pemphaxin, antimitochondrial antibody

## Abstract

**Simple Summary:**

Pemphigus is a group of diseases of the immune system manifesting as blisters localized to the skin and mucosae, such as in the mouth or on the genitals. Pemphigus vulgaris (PV), the most common and severe form of pemphigus, is potentially fatal, and its long-term treatment with corticosteroids and immunosuppressants is associated with significant side effects. Current treatments are not specific to the mechanisms causing the disease. The discovery that PV autoimmunity targets acetylcholine receptors in addition to cell adhesion molecules opened the way to novel pharmacological treatments for this life-threatening disease. In this article, we review the evidence supporting the role of the non-neuronal cholinergic system in PV.

**Abstract:**

The importance of acetylcholine (ACh) in keratinocyte adhesion and acantholysis has been investigated over the last three decades, particularly in the pathophysiology of autoimmune blistering dermatoses. Pemphigus vulgaris (PV) is an autoimmune blistering skin disease where autoantibody-mediated suprabasilar intraepidermal splitting causes flaccid blisters and non-healing erosions of the oral mucosa and sometimes also of the skin. Historically, acantholysis in PV was thought to be driven by anti-desmoglein (Dsg) antibodies. Herein, we describe the role of autoantibodies against keratinocyte muscarinic and nicotinic acetylcholine receptors, as well as the annexin-like molecule pemphaxin that also binds ACh, in the immunopathogenesis of PV. The identification of targets in this disease is important, as they may lead to novel diagnostic and therapeutic options in the future for this potentially deadly disease.

## 1. Introduction

Historically, acetylcholine (ACh) has primarily been viewed as one of the principal neurotransmitters of the central nervous system acting on nicotinic receptors (nAChRs) and muscarinic receptors (mAChRs). More recently, however, ACh production and the expression of its receptors have been shown in a wide variety of cells in different organisms, thus supporting the hypothesis that ACh is an ontogenetically old, universal cytotransmitter, which has only secondarily become specialized in the nervous system [1].

It is now well-established that ACh metabolism plays a vital role in both neuronal and non-neuronal cells, and its importance in keratinocyte function has been elucidated over the last three decades [2]. The highest concentration of free ACh is found in human skin when compared to other organs, such as the lungs, oral mucosa, intestine, and gallbladder. ACh is not only synthesized in human keratinocytes but also secreted as a cytotransmitter and important for keratinocyte vital functions [3]. For example, when human foreskin keratinocytes are treated with ACh receptor (AChR) antagonists, they undergo premature death [2].

The biological effects of ACh in the skin are finely tuned to the regulation of each phase of the cell cycle via the intracellular signaling pathways coupled to each AChR. The regulation of cell–cell and cell–matrix adhesions in keratinocytes is an important biological function of ACh in stratified squamous epithelia. The downstream pathways of ACh are mediated by distinct AChR subtypes, and their targets include both intercellular adhesion molecules, such as classical and desmosomal cadherins, and integrins mediating keratinocyte adhesion to a substrate [4]. It is not surprising, therefore, that the cholinergic system of the skin is involved in blistering diseases that are characterized by keratinocyte separation [5]. The AChRs expressed by epidermal and/or mucosal keratinocytes are targeted by the autoantibodies found in patients with the autoimmune blistering disease pemphigus [6].

The pemphigus group of diseases are defined by autoimmune intraepithelial blistering and composed of four clinical subtypes: pemphigus vulgaris (PV), pemphigus foliaceous (PF), paraneoplastic pemphigus (a.k.a. paraneoplastic autoimmune multiorgan syndrome or PAMS) [7], and IgA pemphigus. Various antibodies to epidermal antigens have been detected in the serum of patients with pemphigus, most commonly immunoglobulin G (IgG). Distinct subclasses of IgG exist, termed IgG1, IgG2, IgG3, and IgG4, and each shows significant differences in their biological activities, most notably in their ability to fix complement and bind to Fc receptors. PV is a potentially lethal form of autoimmune pemphigus, where autoreactive IgG (predominantly the IgG4 subclass) causes suprabasilar intraepidermal splitting, which clinically determines the formation of flaccid blisters and non-healing erosions of the oral mucosa and sometimes also of the skin. Desmogleins (Dsgs) are a group of Ca^2+^-dependent cadherin molecules that bind to desmocollins (Dscs) and provide a connection and structure to adjacent cells. Historically, acantholysis in PV was thought to be driven by autoantibodies against Dsg1 and Dsg3. However, up to 15% of patients with acute PV do not have anti-Dsg1/3 antibodies, determined using ELISA [8]. More recently, autoantibodies against Dsc1 and Dsc3, mitochondrial proteins, human leukocyte antigens, molecules, thyroid peroxidase (TPO), a Ca^2+^/Mn^2+^-ATPase encoded by ATP2C1 (hSPCA1), and several muscarinic and nicotinic AChRs have been identified as novel targets for pemphigus autoimmunity [6,9].

In this review, we discuss the importance of ACh in the regulation of epidermal cohesion and the consequences of the dysregulation of the epidermal cholinergic system in the disruption of cell–cell adhesion.

## 2. Non-Neuronal Acetylcholine in the Skin and Oral Mucosa

The non-neuronal cholinergic system is involved in the physiological regulation of skin cell growth and differentiation, barrier formation, sweat and sebum secretion, microcirculation, and adhesion [10]. In keratinocytes, choline and coenzyme A substrates are metabolized to synthesize ACh. This is the rate-limiting step in ACh de novo synthesis, and ACh is then metabolized by acetylcholinesterase (AChE) [11]. In the epidermis, ACh is produced in all epidermal layers, while AChE is found predominantly in basal cells. Choline acetyltransferase and nicotinic and muscarinic AChRs are also expressed in the oral mucosa and are coupled to the regulation of cell adhesion and motility [12,13].

ACh is a positively charged molecule and does not easily cross the lipophilic cell membrane. Therefore, it achieves its cellular effects by binding to the muscarinic and nicotinic classes of AChRs, which are found on cell membranes and intracellularly. The five muscarinic AChR (mAChR) subtypes (M_1_–M_5_) are composed of a single-subunit transmembrane glycoprotein. The odd-numbered cell membrane mAChRs (M_1_, M_3_, and M_5_) are bound to a pertussis toxin-insensitive G protein, which stimulates phospholipase C and, in turn, releases inositol 1,3,5-triphosphate (IP3), frees Ca^2+^ and diaglycerol from intracellular stores, and activates protein kinases. Conversely, even-numbered mAChR subtypes (M_2_ and M_4_) are bound to a pertussis toxin-sensitive G protein that only weekly stimulates phospholipase C, causing the opening of inward K^+^ channels, the inhibition of adenylyl cyclase, and the augmentation of arachidonic acid release. The various mAChR subtypes are localized to different levels of the epidermis; M_1_ and M_4_ are found in the suprabasal layers, and M_2_, M_3_, and M_5_ are localized in the lower layers [1]. Nicotinic AChRs (nAChRs) are ligand-gated ion channel proteins found on both the surface of keratinocytes and the mitochondrial outer membrane. On the cell membrane, nAChRs facilitate Na^+^ and Ca^2+^ influx and K^+^ efflux. Specifically, human epidermal keratinocytes express the α3, α5, α7, α9, α10, β2, and β4 nAChR subunits [14]. The expressions of nAChRs are highly variable between various body sites and influenced by factors such as age, atopy, smoking history, and prior skin trauma. Additionally, the screening of the skin keratinocyte λgt11cDNA library has identified pemphaxin, also known as annexin 31 or ANXA9, and this also binds Ach [15].

The non-neuronal cholinergic system plays a role in nicotine toxicity in oral keratinocytes and in epithelial cancers [16]. It is now known that nAChRs expressed on the cell membrane and mitochondria mediate both growth-promoting and anti-apoptotic effects synergistically. Other mechanisms associated with nicotine toxicity include the genotoxic action of reactive oxygen species. With regard to mAChRs, experimental results indicate that mAChRs expressed in human keratinocytes regulate various functions, including cell viability, proliferation, migration, adhesion, and terminal differentiation; hair follicle cycling; and the secretion of humectants, cytokines, and growth factors [17]. In the skin and oral tissues, these nicotinic and muscarinic pathways may interact or compete with one another so that the overall biologic effect is a cumulative sum of the individual effects of each AChR expressed by a keratinocyte at a particular stage of its development.

## 3. ACh Role in Keratinocyte Adhesion

In vitro studies have revealed that the complete inhibition of AChRs with mecamylamine, atropine, or strychnine results in a delay in epidermal proliferation and differentiation. The addition of a selective anti-mAChR antibody, atropine, results in a more significant delay in epidermal proliferation and differentiation than the addition of mecamylamine, a selective anti-nAChR antibody, to cell cultures [1]. All treated cells reveal prominent acantholysis in the basal and lower suprabasal layers, leading to keratinocyte death via apoptosis [18]. Contrastingly, treatment with AChR agonists thickened the epithelium [1].

Several AChR subtypes regulate cell-to-cell keratinocyte adhesion. The inhibition of the α3-, α9-, and M3-AChR signaling pathways results in cell–cell detachment and the modulation of the expression levels of E-cadherin and β- and γ-catenins. This has also been confirmed in studies of keratinocytes from AChR knockout mice [1,2]. α3^−/−^ , α9^−/−^, and M3^−/−^ mice have aberrant protein and mRNA keratinocyte expressions of both E-cadherin and β- and γ-catenins. In addition, the stratified epithelium of α3^−/−^ and M3 ^−/−^ mice reveals acantholytic cells and areas of microvesiculation, supporting the role of these receptors in keratinocyte adhesion [4]. At the cellular level, the addition of drugs that activate AChR increases the relative amounts of both classical and desmosomal cadherins in keratinocytes, whereas cholinergic antagonists downregulate the expressions of keratinocyte adhesion molecules [4]. These studies not only elucidated the role of each unique AChR subtype in the cholinergic regulation of keratinocyte adhesion but also revealed the interaction among the different AChR subgroups. These findings reveal that each unique keratinocyte-AChR cooperates in the regulation of adhesion molecules in stratified epithelia, as these receptors are differentially expressed in each stage of keratinocyte development.

## 4. Anti-AChR Autoimmunity in Pemphigus

An abundant body of evidence supports the involvement of the epidermal cholinergic axis in the pathophysiology of pemphigus. Cholinomimetic drugs have been shown to reverse pemphigus-antibody-mediated acantholysis in vitro [19]. Clinically, anti-AChR antibodies have been identified to be present in 85% of patients with pemphigus [20].

It is notable that patients with pemphigus can also develop myasthenia gravis (MG), an autoimmune anti-AChR disease, suggesting a shared immunopathogenic pathway [21].

We have previously reported a patient with both PV and MG successfully treated for five years with an oral acetylcholinesterase inhibitor, Mestinon (pyridostigmine bromide), but the therapeutic role of Mestinon as a non-immunosuppressive treatment option for PV remains to be fully investigated [19].

The roles of specific AChRs and subunits in pemphigus autoimmunity are summarized in Table 1.

In the following paragraphs, we discuss the role of autoantibodies against cholinergic receptors in the disruption of cell–cell adhesion in pemphigus and highlight the pathogenic role of anti-AChR antibodies in blister formation.

### 4.1. Anti-mAChR Autoimmunity in Pemphigus

Electron microscopy imaging reveals that keratinocyte mAChRs are located on cell membranes in association with desmosomes, likely facilitating adhesion between keratinocytes [22]. The treatment of human foreskin keratinocytes with muscarinic agonists resulted in the spread of keratinocyte cytoplasm, increased keratinocyte “crawling-type” locomotion, cell–substrate attachment, and the promotion of intercellular connections [23].

Autoantibodies against mAChRs in patients with PV have been known for many years, but more recently, the role of the M3 subtype of AChRs in pemphigus has been elucidated specifically [24,25,26,27]. M3 mAChR is found predominantly in the basal layer of the epidermis and is involved in the regulation of keratinocyte proliferation, migration, adhesion, and terminal differentiation by endogenous ACh [17]. In mouse models, acantholysis, signified by a positive Nikolskiy sign, can be induced by the synergistic effect of anti-M3 mAChR, anti-desmocollin 3, and anti-secretory pathway Ca^2+^/Mn^2+^-ATPAse isoform 1 protein (SPCA1) autoantibodies [27].

More recently, the long-term incubation of cultured keratinocytes with the anti-M3 mAChR antibody resulted in the decreased expression of M3 mAChR and an agonist-like effect on M3 mAChR signaling [26]. PV IgGs cause acantholysis via a receptor/ligand interaction titled the “Inactivation/Activation of Biologically Active Molecules”, mitigating the participation of complement and inflammatory cells. According to this model, an autoantibody may activate, inactivate, protect, or have no effect on the biological function of the targeted receptor. An interaction with an active site of M3 mAChR may result in a mimetic or agonist-like effect, in imitation of the natural ligand (for example, ACh), leading to morphological changes that are similar to those occurring due to the binding of the natural agonist ACh. Conversely, each of the following three types of M3 mAChR/autoantibody interactions may result in a blocking or an antagonist-like effect, thus inactivating ACh signaling: (i) the physical blocking of the ligand-binding site so that ACh is unable to bind or binds with diminished avidity; (ii) the antibody/receptor complex is internalized or degraded, decreasing the number of functional M3-mAChRs on the cell surface; and (iii) alteration or allosteric regulation that influences the ligand-binding ability of M3AR. In one example, short-term exposures of cultured keratinocytes to the muscarinic agonist muscarine, PV IgG, or both led to changes in the expressions of keratins 5 and 10, consistent with the inhibition of proliferation and the upregulation of differentiation pathways, consistent with the biological function of M3 mAChR [26]. In contrast, long-term incubations induced a keratin expression pattern suggestive of upregulated proliferation and decreased differentiation, in keeping with the hyperproliferative state of keratinocytes in PV. This change is likely the effect of the desensitization of M3 mAChR, representing the net antagonist-like effect of PV IgG. Thus, the degradation of M3 mAChR that disrupts the normal folding of keratinocytes in the epidermis of patients with PV resulting in acantholysis represents an alternative pathway of acantholysis in patients with anti-Dsg1/3-negative PV. In these patients, autoantibodies against M3 mAChR cause an intraepidermal split and, in association with anti-SPCA1, cause acantholysis. In one study, 45 patients with active pemphigus had a positive correlation of anti-M3 mAChR autoantibody titers with disease severity at baseline, 3 months, and 15 months, and the titers declined with therapy [28]. That study also identified that anti-M3 AChR autoantibody titers were higher in patients with PV than in patients with PF.

M3 mAChR knockout mice (*Chmr3*^−/−^ are noted to have an altered intercellular cohesion of basal cells, epidermal hyperplasia, and modulation of genes that contribute to intercellular adhesion and cell arrangement [29]. Notably, *Chmr3*^−/−^ mice and patients with PV both display an altered epidermal morphology [29]. Most recently, our team characterized the cellular and molecular mechanisms by which M3 mAChR controls the epidermal structure and function. In this study, we identified differentially expressed genes in specific subpopulations of epidermal cells in neonatal mice deficient in M3 mAChR using single-cell (sc) RNA-seq. *Chmr3*^−/−^ mice notably featured an abnormal epidermal morphology, such as the shrinkage of basal keratinocytes, the proliferation of nucleated basal cells, and the augmentation of intercellular spaces. These morphologic changes were paralleled by increased cell proliferation genes and a decrease in genes contributing to epidermal differentiation, extracellular matrix formation, intercellular adhesion, and cell arrangement. These findings provide new insights into how ACh controls epidermal differentiation, and they lay the foundation for future translational studies evaluating the therapeutic potential of cholinergic drugs in dermatology.

### 4.2. Anti-nAChR Autoimmunity in Pemphigus

nAChRs are found not only on the cell surface but also on the mitochondrial outer membrane [30]. Mitochondrial nAChRs control cytochrome c (CytC) release by blocking mitochondrial permeability transition pore (mPTP) opening, thus preventing mitochondrial apoptosis. Therefore, nAChRs play a key role in maintaining keratinocyte viability [31].

The binding of PV IgGs that antagonize mitochondrial nAChRs results in the opening of the mPTP; the release of CytC; caspase-9 activation; and the induction of apoptosis cascades [32,33,34,35], which represents a unique mechanism of keratinocyte damage in pemphigus termed apoptolysis [36]. When keratinocytes were cultured with α3-nAChR antagonists, such as mecamylamine and κ-bungarotoxin, acantholysis resulted [37]. When compared to normal subjects, patients with PV and PF were found to have increased anti-nAChR autoantibodies [36]. In that study, the correlation of IgG autoantibodies against nAChRs and Dsg molecules was reported in patients with pemphigus treated with steroid-sparing agents or rituximab. In another study, all patients with PV with a positive anti-γ/ε nAChR serology had mucocutaneous involvement, and there was a positive correlation between this autoantibody value and disease severity at baseline [38]. Interestingly, the anti-γ/ε nAChR titers did not correlate with the anti-Dsg1/3 values, supporting the view that anti-nAChR autoantibodies act independently of anti-Dsg1/3 autoantibodies and are present.

**Table 1 biology-12-00354-t001:** Pathogenic anti-AChR antibodies implicated in pemphigus.

AChR Targeted	Role of AChRs in Autoimmune Blistering Disease	Reference
M3 mAChR	Autoantibodies cause an intraepidermal split and acantholysis in patients with anti-Dsg1/3 autoantibody-negative PV.	[24]
M3 mAChR	Positive correlation of anti-M3 mAChR autoantibody titers with disease severity and the titers declined with therapy (*n* = 45). Anti-M3 mAChR autoantibody titers are higher in patients with PV than in patients with PF.	[29]
M3 mAChR	Long-term exposure of anti-M3AChR in cultured keratinocytes has an agonist-like effect on M3 mAChR signaling, disrupting the normal folding of keratinocytes in the epidermis and resulting in acantholysis.	[26]
M3 mAChR	M3 mAChR knockout mice are noted to have altered intercellular cohesion of basal cells, epidermal hyperplasia, and modulation of genes that contribute to intercellular adhesion and cell arrangement.	[28]
α3 nAChR	Keratinocytes cultured with α3 nAChR antagonists undergo acantholysis.	[35]
nAChR	Patients with PV and PF, when compared to normal subjects, have increased anti-nAChR autoantibodies.	[37]
γ/ε nAChR subunits	Positive anti-γ/ε nAChR serology is associated with mucocutaneous involvement, and there is a positive correlation between this autoantibody value and disease severity at baseline.	[38]
α9 nAChR	Addition of anti-α9 nAChR antibody causes keratinocyte acantholysis, which is reversed with a cholinergic agonist.	[39]
α3, α5, α7, α9, α10, β2, and β4 nAChR subunits	PV IgGs precipitate mitochondrial nAChR subunits.	[21]
Pemphaxin (PX)	Anti-PX PV alone does not cause acantholysis, but addition of anti-PX to PV IgG restores acantholytic activity.	[15]

Based on experimental data, the key AChRs and subunits involved in the pathophysiology of PV appear to be α7, α9, and mitochondrial nAChRs. An analysis of pooled disease-causing PV IgGs revealed the role of human α9-AChR in PV. In vivo, the addition of the anti-α9 AChR antibody caused keratinocyte acantholysis, which was reversed with the cholinergic agonist carbachol [39]. Interestingly, α9-AChR can be blocked by both nicotinic and muscarinic antagonists and is one of the few AChRs that has dual muscarinic and nicotinic pharmacology.

### 4.3. Other AChRs Targeted in Pemphigus

The screening of keratinocyte cDNA revealed a new “human annexin-like molecule”, named pemphaxin (PX), which has been found to act as a dual muscarinic and nicotinic cholinergic receptor [15]. In animal models, the removal of anti-PX autoantibodies from PV sera eliminated acantholytic activity and eluted antibody immunoprecipitated native PX. Anti-PX PV alone, however, did not cause acantholysis, but the addition of anti-PX to PV IgG restored acantholytic activity, suggesting its role in skin blistering in patients with PV [15]. The annexin family of molecules is involved in glucocorticoid anti-inflammatory signaling in the skin, but the elucidation of the specific role of PX in this pathway is still pending.

### 4.4. Role of AChRs in the Multipathogenic Theory of Pemphigus Pathophysiology

Numerous studies have shown that, rather than being monopathogenic, the PV IgG fraction contains an array of autoantibodies against different keratinocyte antigens evoking distinct intracellular mechanisms [40], and that PV IgG targeting non-Dsg antigens are pathogenic [8]. An in vitro pathogenicity analysis of the top 10 non-Dsg PV autoantibodies identified using the proteomic technique [41] revealed the abilities of anti-M3 mAChR, the anti-ε nAChR subunit, anti-Dsc3, and anti-SPCA1 IgGs to induce the dissociation of the keratinocyte monolayer and release CytC (Table 2).

These observations have led to the formulation of a multipathogenic theory of pemphigus pathophysiology, which explains intraepidermal blistering through the “multiple hit” hypothesis. The major premise for a multipathogenic theory is that the signaling pathways elicited individually by each autoantibody can be overcome, as each single type of autoantibody only induces reversible changes, and affected keratinocytes can recover due to self-repair. When the salvage pathway and/or other cell functions are altered by a partnering autoantibody and/or other pathogenic serum factors, the damage then becomes irreversible. According to this hypothesis, a simultaneous and synchronized inactivation of the physiological mechanisms regulating keratinocyte cell–cell adhesion is necessary to disrupt epidermal integrity. Individual variations within pathogenic PV-IgG targeting different keratinocyte proteins likely determine the magnitude of the “multiple hit” attack required to disrupt epidermal integrity in a particular patient with PV and explain the clinical and immunopathological variability of PV. The key autoantibodies include not only PV IgG targeting adhesion molecules and AChRs but also autoantibodies targeting the inner and outer mitochondrial membrane and matrix. Antimitochondrial antibodies (AMA) and anti-Dsg1/3 autoantibodies are thought to act synergistically to induce acantholysis in mouse skin.

For example, the removal of AMA abolishes the acantholytic effect of PV serum, but the treatment of cells with AMA alone does not result in acantholysis; however, the treatment of these cells with a mixture of AMA and anti-Dsg antibodies induced acantholysis [31]. Different acantholytic factors are amenable to specific treatment strategies; therefore, it is not surprising that mitochondrion-protecting drugs, such as tetracyclines and niacinamide, are used in the potentially curative treatment of PV [42].

## 5. Therapeutic Implications of Elucidation of Anti-AChR Autoimmunity in Pemphigus

The pathogenesis of pemphigus is most adequately explained through a “multiple hit hypothesis”, wherein acantholysis is mediated by the synergistic effects of anti-AChR autoantibodies and antibodies directed to adhesion molecules, such as Dsgs and Dscs. The presence of distinct classes of autoantibodies in patients with PV determines the activation of specific signaling pathways, which, in turn, may lead to the development of different clinical phenotypes and possibly account for a variable response to treatment. For example, whilst the pathophysiology of anti-Dsg pemphigus is dominated by the activation of p38MAPK and is associated with desmosome disassembly, anti-Dsc3 and anti-SPCA1 AuAbs activate SRC proto-oncogene, cytochrome c release, and caspase-9 activity in patients with anti-Dsg-negative PV [27]. Likewise, the anti-M3 antibody determines the level of intraepidermal split and acts synergistically with other non-Dsg IgGs in patients with non-Dsg PV [24]. It is also worth mentioning that PV pathophysiology involves non-IgG-mediated signaling via the FAS ligand [43] and PERK [44], which link endoplasmic reticulum stress, mitochondrial dysfunction, and apoptosis. Intriguingly, mutations of SPCA1 are associated with acantholytic skin disorders, and this supports the notion that ER stress and alterations in intracellular calcium are associated with acantholysis. The complexity of PV pathophysiology may explain the variability in treatment response and the unsuccessful attempts to find a mechanism-based pharmacological treatment for PV to date [45].

Autoimmunity in pemphigus is directed against multiple organ-specific and non-organ-specific proteins, some of which are also targeted in other types of autoimmune diseases. For example, autoantibodies targeting muscle AChRs are detected in MG, an autoimmune disease characterized by muscle weakness and the fatiguability of skeletal muscles [46].

The finding of anti-AChR IgG is also common in patients with paraneoplastic pemphigus, and a sizeable number of these patients experience muscle weakness [47]. Targeting the cholinergic pathways can serve as a potential novel therapy for PV. Treatment with Mestinon (360 mg/d) in a patient with PV that was resistant to treatment with systemic glucocorticosteroids allowed him to keep his disease under control at a lower dose of prednisone than he had used before starting Mestinon treatment [48]. A double-blinded placebo-controlled study evaluated the effectiveness of topical pilocarpine 4% gel, a cholinomimetic agent, versus placebo in three patients with PV and noted that re-epithelialization was significantly higher in the treatment group [49]. Cigarette smoking, which can activate keratinocyte nAChRs, has been reported to improve pemphigus [50]. Indeed, there is a reciprocal correlation with pemphigus disease activity. A case–control study of patients with pemphigus vulgaris revealed that 25.9% of 126 patients were smokers versus 48.5% of controls [51]. A case–control study found a lower proportion of current or former smokers in patients than in controls: 15.3% of 59 patients compared with 47.4% of the general population [52]. In a study of 210 patients with pemphigus (199 patients with pemphigus vulgaris and 11 with pemphigus foliaceous) and 205 control subjects, significantly fewer patients in the pemphigus group (17.1%) reported a current or past history of smoking than those in the control group (27.3% smokers). The duration of smoking and the number of cigarettes smoked daily were also significantly lower in the patients with pemphigus [53]. Smokers with PV achieve partial remission more frequently than non-smokers at the end of the 1st year of treatment. Furthermore, the number of patients with PV in remission at the end of the 2nd year of therapy is significantly higher for smokers than for non-smokers [54]. As the role of ACh in pemphigus is further elucidated, cholinomimetics may play a larger role in providing another corticosteroid-sparing option for this potentially deadly disease. The elucidation of immunopharmacologic aspects of autoimmunity against keratinocyte AChRs should open the door for the evaluation of cholinergic drugs for the steroid-sparing therapy of patients with some other diseases of skin adhesion featuring acantholysis.

## Figures and Tables

**Table 2 biology-12-00354-t002:** Pathogenicity of top 10 non-Dsg autoantibodies immunoaffinity-purified from patients with acute PV.

Gene Symbol of the Targeted Protein	Name of the Self-Antigen	% of All Patient Sera	% of Anti-Dsg3 Negative Sera	Keratinocyte Dissociation Score	Cytochrome c Release Activity
*HLA-DRA*	major histocompatibility complex, class II, DR α chain	45	58	−	−
*DSC3*	desmocollin 3 (Dsc3)	44	54	+ (2/2) *	−
*DSC1*	desmocollin 1 (Dsc1)	44	53	−	−
*ATP2C1*	human secretory pathway Ca^2+^/Mn^2+^ATPase protein 1 (SPCA1)	43	59	+ (3/3)	+ (3/3)
*PKP3*	plakophilin 3	43	52	−	−
*CHRM3*	M_3_ muscarinic acetylcholine receptor	42	54	+ (1/2)	−
*COL21A1*	collagen, type XXI, α1	42	51	−	−
*ANXA8L1*	annexin A8-like 1	42	50	−	−
*CD88*	complement component 5a receptor 1	42	50	−	−
*CHRNE*	ε subunit of nicotinic acetylcholine receptor	41	53	+ (2/2)	+ (2/2)

* Number of positive results produced by autoantibodies from individual patients with PV over the total number of tested patients with PV. Plus (+) indicates statistical significance (*p* < 0.05) compared to the effect of equivalent concentration of normal IgG (control), and minus (−) indicates no significant changes (*p* > 0.05). Each experiment was performed in triplicate.

## Data Availability

No new data were created for this manuscript.

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
