# Peer review of "The Role of Non-Neuronal Acetylcholine in the Autoimmune Blistering Disease Pemphigus Vulgaris"

_biology, 2023, doi:10.3390/biology12030354_

Round 1
Reviewer 1 Report
The review by Foulad et al provides a current and comprehensive overview of the non-neuronal acetylcholine and cognate receptors in skin and epidermal physiology and pathology. Grando and colleagues made the initial discoveries and established the critical roles of this ligand-receptor system in keratinocyte differentiation, stratification, acantholysis and pemphigus.
Recent studies have further delineated and elaborated on the interrelationship of desmoglein and non-desmoglein antibodies in pemphigus disease activity and pathogenesis.
In this regard, could the authors further elaborate or speculate on molecular differentiation of different clinical types or response to various therapies based on the clinical types of pemphigus and those patients with non-desmoglein pemphigus?
Are there other skin diseases that show acantholysis other than pemphigus that may be regulated or mediated by the ACh ligand-receptor system?
Overall, the review is very well written and provides an up-to-date summary of this novel research field and its clinical correlates for pemphigus.
Author Response
please find attached our point-by-point response

Reviewer 2 Report
AAs presented a comprehensive review on the role of non-neuronal acetylcholine in pemphigus. The review is well written and well referenced. This work will enlarge the knowledge on the importance of non-desmoglein auto-antibodies present in pemphigus.
The paper can be accepted without any further changes.
Author Response
Many thanks for your positive comments.
Reviewer 3 Report
SUMMARY
Foulad et al present a very well-written manuscript on the role of different classes of autoantibodies targeting Ach or its receptors in the pathogenesis of pemphigus. The main body is well structured and has a thorough concept by describing in detail how autoantibodies induce skin blisters. The discussion of different effector mechanisms by autoantibodies is interesting and well-written.
However, the authors did not differentiate between autoantibody classes which is a bit peculiar. Especially in light of the newly emerging concept of autoreactive IgM as disease driver and also pathology inducer. In particular, recent literature also describes a critical role for autoreactive IgM in the pathogenesis of pemphigus. This needs to be discussed extensively in the manuscript in order to make it a valuable addition to the field.
MAJOR COMMENTS
- In the introduction the authors describe that 15 % of PV patients do not show anti Dsg autoantibodies (via ELISA). Mostly, autoantibodies refer to IgG class antibodies. However, IgM autoantibodies inducing pathology are well characterized nowadays. (see Hirano, Y.; Iwata, H.; Tsujuwaki, M.; Mai, S.; Mai, Y.; Imafuku, K.; Izumi, K.; Koga, H.; Ujiie, H. Super-resolution imaging detects BP180 autoantigen in immunoglobulin M pemphigoid. J. Dermatol. 2022, 49, 374–378, doi:10.1111/1346-8138.16260.). It is very well conceivable that IgM autoantibodies (primary antibodies) are the first line in establishing a full-blown autoimmune disease. Please discuss that later on (notably, it is important to not confuse natural IgM (which recognizes harmful altered self-structures) with actually autoreactive IgM recognizing protein antigens leading to pathology!).
- A general paragraph mentioning different antibody classes with small explanations (basic facts) would be good for the overall understanding of the text by readers (in Introduction).
- Regarding the anti AChR autoantibodies in pemphigus, it would be interesting to discuss this matter with keeping myasthenia gravis in mind (https://doi.org/10.3389/fimmu.2020.00212). MG is also characterized by anti-AChR autoantibodies. Might there be a common origin of autoreactive B cells? How are these B cells activated? Do these autoantibodies (MG vs PV) share the same epitope or affinity? Why are symptoms different (effectors etc.)? This would add more depth to the discussion.
- Since the authors discuss interesting data on how different autoantibodies induce pathology in PV, it would be a great addition to add a graphical summary of the autoantibody table to visualize autoantibody effector functions (that cause pathology).
- Line 178 states that the ligand might bind with reduced affinity when autoantibodies are present. Do the authors mean avidity of the interaction? Meaning that the ligand binds less stable and therefore dissociates quicker? Is there BLI or PSR data available testing this hypothesis?
- Line 199 stating interesting findings by the authors lacks a citation.
- Line 216 is hard to follow. Do the autoantibodies bind to mitochondria that are already exposed by incomplete apoptosis? Please explain this more.
- Multiple hit theory needs more explanation. Is it required to do multiple hits in regard of effector mechanism or epitopes/receptors targeted?
Author Response

(The authors gave the same response as above.)

Round 2
Reviewer 3 Report
The authors addressed all concerns and I can promote publication at this stage.
The article is a very valuble addition and interesting.